# Mechano-Chemical Properties of Electron Beam Irradiated Polyetheretherketone

**DOI:** 10.3390/polym14153067

**Published:** 2022-07-29

**Authors:** Nurlan Almas, Bayan Kurbanova, Nurkhat Zhakiyev, Baurzhan Rakhadilov, Zhuldyz Sagdoldina, Gaukhar Andybayeva, Nurzhan Serik, Zhanna Alsar, Zhandos Utegulov, Zinetula Insepov

**Affiliations:** 1Department of Science and Innovation, Astana IT University, Nur-Sultan 010000, Kazakhstan; n.almas@astanait.edu.kz (N.A.); nurkhat.zhakiyev@astanait.edu.kz (N.Z.); 2Department of Physics, School of Sciences and Humanities, Nazarbayev University, Nur-Sultan 010000, Kazakhstan; bayan.kurbanova@nu.edu.kz; 3Surface Engineering Tribology Centre, Amanzholov University, Oskemen 070020, Kazakhstan; brahadilov@vku.edu.kz (B.R.); zhsagdoldina@vku.edu.kz (Z.S.); gaukharandybayeva@gmail.com (G.A.); 4Department of Engineering, Astana International University, Nur-Sultan 010000, Kazakhstan; serik.nurzhan@list.ru (N.S.); j.alsar.979@gmail.com (Z.A.); 5School of Nuclear Engineering, Purdue University, West Lafayette, IN 47907, USA; 6Department of Condensed Matter Physics, National Research Nuclear University MEPhI, 115409 Moscow, Russia

**Keywords:** polyetheretherketone, electron beam irradiation, nanoindentation, roughness, hardness, Young’s modulus, elastic constant

## Abstract

In this study, the mechano-chemical properties of aromatic polymer polyetheretherketone (PEEK) samples, irradiated by high energy electrons at 200 and 400 kGy doses, were investigated by Nanoindentation, Brillouin light scattering spectroscopy and Fourier-transform infrared spectroscopy (FTIR). Irradiating electrons penetrated down to a 5 mm depth inside the polymer, as shown numerically by the monte CArlo SImulation of electroN trajectory in sOlids (CASINO) method. The irradiation of PEEK samples at 200 kGy caused the enhancement of surface roughness by almost threefold. However, an increase in the irradiation dose to 400 kGy led to a decrease in the surface roughness of the sample. Most likely, this was due to the processes of erosion and melting of the sample surface induced by high dosage irradiation. It was found that electron irradiation led to a decrease of the elastic constant *C*_11_, as well as a slight decrease in the sample’s hardness, while the Young’s elastic modulus decrease was more noticeable. An intrinsic bulk property of PEEK is less radiation resistance than at its surface. The proportionality constant of Young’s modulus to indentation hardness for the pristine and irradiated samples were 0.039 and 0.038, respectively. In addition, a quasi-linear relationship between hardness and Young’s modulus was observed. The degradation of the polymer’s mechanical properties was attributed to electron irradiation-induced processes involving scission of macromolecular chains.

## 1. Introduction

PEEK (polyetheretherketone), originally developed by Imperial Chemical Industries in 1978, is a semi-crystalline thermoplastic with unique chemical, mechanical and electrical properties [1]. This polymer possesses excellent mechanical, thermal and electrical properties, and is resistant to harsh external stimuli such as radiation [2], chemically active environments [3], pressure, temperature and others [4]. Thanks to these properties, the polymer is widely used in the aerospace [5,6] and automotive industries [7] for the manufacture of doors, pipes for protecting high-voltage cables, thermal acoustic blankets, brackets, pump gears, and valve seats and other constructive elements. In the nuclear industry, the polymer is proposed to be used for containers for long-term storage of spent nuclear fuel and high-level radioactive waste [8,9,10]. In biomedical applications, PEEK is considered as an implant alternative to traditional metal and ceramic materials, for example, in the fields of dentistry [11] and orthopedics [12]. The development of porous PEEK-based membranes for such applications as lithium-ion batteries [13,14], gas separation [15], organic solvents nanofiltration [16], etc., is considered very promising.

Polymers are often used in harsh radiation environments, and radiation-induced effects drastically change their structure and properties [17]. The degradation of the mechanical properties of PEEK, polyamide, polyether-imide, polyether-sulfone, and polysulfone polymers upon irradiation with electrons at a dose of up to 100 MGy was reported [18]. According to the literature, PEEK showed the highest radiation resistance among these listed polymers. Recently, PEEK has been studied with respect to its mechanical properties when irradiated by 20 MeV $H^{2+}$ ions with the dose of 14.5 MGy, and showed an abrupt decrease in tensile strength [19]. However, 1 MeV $H^{+}$ ion irradiation with the fluence of 3 × $10^{9}$ ions/$m^{2}$ leads to an increase of the hardness value [20]. The same trend of hardness change was reported after UV irradiation with the energy fluence of 3.47 × $10^{4}$ ions/$m^{2}$ [21]. A reduction in Young’s modulus was observed after irradiation by gamma rays at 600 kGy doses [22]. As the main interactions with matter are basically the same for gamma rays and high-energy electrons [23], it is essential to compare the electron irradiation effects on polymers. However, a limited number of works are devoted to the study of the effects of high energy electron radiation on the structure and properties of polymers [24,25]. The authors of [26] reported that high-density polyethylene (HDPE) and low-density polyethylene (LDPE) polymers exposed to 198 kGy exhibited hardness 57% and 24% higher, respectively, than non-irradiated materials. At the same time, the elastic modulus of the HDPE polymer, irradiated at a dose of 198 kGy, increased by 38%, and for the LDPE irradiated at a dose of 132 kGy, the elastic modulus increased by 21%. The breaking strength of PEEK reduces sharply and yield strength reduces slightly due to the 1 MeV electron irradiation with the fluences of 5 × $10^{14}$ e/${\mathrm{cm}}^{2}$ and 3 × $10^{15}$ e/${\mathrm{cm}}^{2}$ [27]. High energy electrons with the energy of 350 keV and doses of 12–34 MGy induces chemical modifications of PEEK’s macromolecules, consequently, both cross-linking, chain scission and a slight decrease of the modulus were observed by differential scanning calorimetry (DSC) and dynamic mechanical analysis (DMA) [28]. However, Brillouin spectroscopy is a non-destructive, rapid and very accurate technique that allows the probing of micro or sub-nano scale perturbations in the system with the comparison of these conventional low-frequency techniques, with certain limitations regarding the measurement resolution [29].

To the best of our knowledge, the mechanical properties of PEEK after 2.7 MeV electron irradiation with doses of up to 400 kGy have never been studied, since the majority of works have been provided on the MGy doses. The purpose of this work is to study the effect of high-energy electron beam radiation with the doses of 200 kGy and 400 kGy on the structure and mechanical properties of the PEEK polymer. In this regard, the morphology of the pristine and irradiated PEEK samples, irradiated at a maximum dose of 400 kGy, were determined by the scanning probe microscopy (SPM), while their elastic properties were assessed by contact-based nanoindentation and temperature-dependent optical (non-contact-based) Brillouin light scattering spectroscopy. Chemical variation due to electron irradiation was determined by Fourier transformed infrared spectroscopy (FTIR). In addition, the depth profiling of electrons penetrating through the PEEK structure was demonstrated by CASINO simulation.

## 2. Materials and Methods

### 2.1. Electron Irradiation

Polyetheretherketone (PEEK) plates, with an area of 10 mm^2^ and 5 mm thickness, manufactured by Ensinger (based on Victrex^®^ PEEK 450 G or Solvay‘s KetaSpire^®^ KT-820 polymer) were investigated. The polymer plates underwent irradiation by fast electrons at room temperature using the ILU-10 industrial electron pulse accelerator facility at the Park of Nuclear Technologies in Kurchatov, Kazakhstan [30]. An alternating high-frequency (HF) electric field was used to accelerate electrons. The radiation source was a copper toroidal resonator equipped with a triode electron gun operating at 50 MHz frequency. The accelerator’s electron beam energy and beam current were set at 2.7 MeV and 6.87 mA, respectively. The electron beam diameter on the sample surface and pulse duration were set to 10 mm and 0.4−0.5 ms, respectively. The polymer plates were exposed to a total dose of 400 kGy.

### 2.2. SPM

The surface morphology of the PEEK samples was measured using the *Smart SPM1000* SPM. AppNano silicon probes, ACST series, designed for soft tapping/non-contact mode measurements were used. A cantilever with the size of 150 μm × 28 μm × 3 µm and stiffness 7.8 N/m was oscillated at the frequency of 150 kHz. The tetrahedrally shaped, Au coated, 14–16 μm long cantilever tip was made of n-type antimony doped single crystalline Si with resistivity 0.01–0.025 ohm/cm and 6 nm radius of curvature.

### 2.3. Nanoindentation

All PEEK samples were nanoindented using the Bruker Hysitron TI Premier nano-scale imaging and surface analysis system. Young’s modulus and hardness were measured using the Oliver and Farr method [31]. The penetrating body was a Berkovich indenter (Ѳ = 65.27°) with the maximum load of 10 mN, and holding time of 5 s. Nanoindentation was tested from 100 points and results were averaged for each sample.

Indentation hardness *H* was determined using the following equation:(1)$$H=\frac{P_{max}}{A_{c}},$$
where *P_max_* is the maximum applied load; *A_c_* is the contact area defined by
(2)$$A_{c}=24.5\;h_{c}^{2}$$
where $h_{c}$ is the indentation contact depth as follows:(3)$$h_{c}=h_{max}-\varepsilon \frac{P_{max}}{S},$$
with *h_max_* and *S* = *dP*/*dh* being the maximum contact indentation depth and contact stiffness or unloading slope, respectively. The *ε* is a constant depending on the indenter shape (*ɛ* = 0.75 [31]). *P_max_*, *h_max_*, and *S* were all obtained from measurements.

Young’s elastic modulus was given by the following formula:(4)$$E=\frac{dP}{dh}\;\frac{1}{2}\frac{\sqrt{\pi }}{\sqrt{A_{c}}},$$
where *P* is the applied load and *h* is the indentation contact depth [31].

### 2.4. Brillouin Spectroscopy and FTIR

Brillouin spectra of the initial and irradiated polymer plates were recorded in the 180° backscattering geometry using a (3+3)-pass tandem JRS Fabry-Perot interferometer [32] (Table Stable Inc., Mettmenstetten, Switzerland). The free spectral range was set at 25 GHz. The incident 532 nm wavelength laser light was focused down to 2 µm spot diameter using 20× microscope objective, while the optical power of the laser beam was kept below 10 mW to prevent sample damage. During optical (non-contact) Brillouin measurements, the sample temperature was controlled using a Peltier PE120 temperature-controlled stage (Linkam Instruments, Inc., Redhill, UK) over a 20–120 °C temperature range. Brillouin spectra were measured from 5 different points for each temperature value and for each sample.

FTIR spectra of the samples were measured using a Thermo Scientific (Waltham, MA, USA) Nicolet iS10 spectrometer. The spectrometer was equipped with automatic spectral identification tools.

## 3. Results

### 3.1. Assessment of Electron Penetration Depth

In this study, monte CArlo SImulation of electroN trajectory in sOlids (CASINO) software was used to estimate the penetration depth of electrons, backscattered and transmitted energy in PEEK samples. The complete electron trajectories in the material was estimated using Monte-Carlo simulation, as described in detail by Drouin et al. [33,34]. PEEK was modeled using the chemical composition density of 1.3 $\frac{g}{{\mathrm{cm}}^{3}}$ and thickness of 5 mm, as given by the supplier. The number of electrons was 100,000 and the specified electron beam diameter was 10 mm, where the irradiation covered the whole sample surface. Figure 1 represents the simulation results of PEEK irradiated with 2.7 MeV electron energy. It can be clearly seen that the electrons with the energy of 2.7 MeV completely pass through the sample (Figure 1a). When the energy of the primary beam increases, the penetration depth is also enhanced [35]. The depth profiling of electron distributions in the sample is shown in Figure 1b. The number of electrons inside the sample is much lower than the initial electron number, and this indicates that the 2.7 MeV electron beam with such strong energy can completely penetrate the entire PEEK structure, and that less electron detainment occurred in the sample. It is also seen in Figure 1c, where most of the electrons are transmitted rather than absorbed in the 2.7 MeV irradiated sample, and their energies are ~1.5 MeV. The electrons continuously change their direction of travel across the polymer due to their elastic backscattering with the host atomic nuclei [36]. The backscattered electrons reach a maximum energy of 2.1 MeV during incident, and 2.7 MeV electron beams, respectively, as displayed in Figure 1d. Moreover, the estimation of electron penetration depth gives valuable information to make sure that we have provided all measurements exactly where the electrons passed and made structural changes. Simultaneous cross-linking and chains scission of PEEK macromolecules under high-dose electron beam irradiation was observed in Ref. [37].

The 2D and 3D morphology images of pristine (a, b) and irradiated (c–f) PEEK are shown in Figure 2.

The roughness analysis of pristine and irradiated PEEK was performed for a scan area of 101.841 µm^2^ and 101.642 µm^2^, respectively. The sample surface before irradiation (Figure 2a) exhibited irregular grooves and bumps. Apparently, these grooves were formed as a result of polishing the sample. The average surface roughness (Ra) of pristine PEEK was relatively high at 265 nm (See Table 1). The irradiation of PEEK with a dose of 200 Gr led to a significant increase in the number of bumps. Wherein, the bumps’ length decreased, their depth increased. The average surface roughness (Ra) of the irradiated PEEK increased significantly and amounted to 750 nm. We believe that the increase in the surface roughness of the sample was caused by radiation damage during irradiation. With an increase in the irradiation dose to 400 kGy, the surface of the sample was smoothed, and the average surface roughness decreased to 665 nm. Most likely, this was due to evaporation and partial melting of the sample surface.

### 3.2. Nanoindentation

Typical loading and unloading curves for pristine and irradiated PEEK surfaces are shown in Figure 3. The loading-unloading speed was set at 200 mN/s with the penetration depth reaching a maximum of 1100 nm. For each sample, 100 indents were implemented. The indents were located at a distance of about 5–10 µm from each other. The trapezoidal load function was such that loading, holding and unloading segments lasted for 5, 10 and 5 s, respectively. In order to minimize the influence of the sample surface roughness on the indentation result, the smoothest areas were selected. The atomic force microscopy (AFM) images of the sample surface subjected to indentation are shown in Figure 4.

The indentation hardness and elastic modulus data as a function of indentation depth for the pristine and irradiated PEEK samples are shown in Figure 5a,b respectively.

The results of measuring the hardness and elastic modulus of PEEK samples before and after electron irradiation by the nanoindentation method are presented in Table 2.

The proportionality constant between *H* and *E* values for pristine PEEK was 0.039, and results for PEEK samples irradiated at 200 kGy and 400 kGy were the same, being 0.038. Our results are consistent with the literature data as shown in Table 3. Besides, *H*/*E* values measured for a range of materials using nanoindentation vary from 0.0067 to 0.1, and our *H*/*E* results are in this range [38].

A close relationship between such properties of polymers as the elastic modulus (*E*), shear modulus (*G*), yield strength (*Y*) and hardness (*H*) was shown in the work [39]. Tabor [40] found that *H* is proportional to the macroscopic yield stress as follows:(5)H≈3Y

Another approximate formula for amorphous polymers was derived by Struik [41]:(6)E≈30Y

Combining Equations (5) and (6) we can derive a very simplified formula as follows:(7)H≈3Y≈E/10

Although the values of the constants in Equation (7) are approximate, the linear relationships between *H*, *Y*, *E* and G usually hold very well. In particular, linear H-Y-E-G ratios were observed for amorphous and semicrystalline polymers [39,42] and for cross-linked polymers [43,44]. Moreover, linear proportionality between properties has also been observed for multiphase, multicomponent polymer systems [45,46].

### 3.3. Brillouin Specrtoscopy

Stokes and anti-Stokes sides of the Brillouin spectra of pristine and irradiated PEEK samples under different temperatures are shown in Figure 6. The signal acquisition time was 10 min for each spectrum. All reported positions of measured Brillouin peaks are the results of the Voigt fitting.

Figure 7 shows the frequency shift of longitudinal acoustic (LA) phonons from pristine and irradiated PEEK samples measured at different temperatures.

The connection between the frequency shift of *ϑ* of Brillouin peaks and the phase velocity *V* of the corresponding acoustic mode directly derived as follows [47,48]:(8)$$V_{L}=\frac{\lambda_{i}\vartheta_{L}}{2n},$$
where *ϑ_L_* is the longitudinal acoustic phonon frequency deduced from the Brillouin peak frequency shift; *λ_i_* is the incident light wavelength and *n* is the refractive index of the material.

Using measured LA phonon velocity one can estimate longitudinal elastic modulus *C*_11_ [48,49,50] as follows:(9)$$C_{11}=\rho V_{L}^{2},$$
where *ρ* is the density of the material. We observe the softening of LA phonon velocity resulting in the subsequent decrease of $C_{11}$ with the rise of temperature, assuming that the PEEK’s mass density was not altered.

As the backscattering geometry only allows the frequency of LA phonons to register, the frequency of transverse acoustic (TA) phonons was taken from the literature [48] to assess TA phonon velocity using modified Formula 8, according to the scattering geometry. Consequently, assuming that PEEK is elastically isotropic, the shear elastic modulus $C_{44}$ can be determined as follows:(10)$$\begin{array}{c} C_{44}=\rho V_{b}^{2},\\ V_{b}^{2}=V_{L}^{2}-\frac{4}{3}V_{T}^{2} \end{array}$$
where $V_{b}$—bulk sound velocity.

Furthermore, the knowledge of compressional and shear modules allows us to find Young’s modulus as follows:(11)$$E=\frac{C_{44}\left(3C_{11}-4C_{44}\right)}{C_{11}-C_{44}}$$

Figure 8 shows the variation of LA phonon velocity (a) and *C*_11_ elastic module (b) with temperature. Finally, in Table 4 we list comparative results of Young’s moduli determined by nanoindentation and Brillouin spectroscopy for pristine PEEK, but it was not possible to compare Young’s moduli for irradiated PEEK due to the lack of transversal phonon values available from the literature. However, as can be seen from Table 4, Young’s moduli values found using both techniques are quite near to each other for pristine PEEK. It proves that LA sound velocity and $C_{11}$ elastic constant values are correct for the irradiated sample.

### 3.4. FTIR

The FTIR spectra of pristine and irradiated PEEK samples are shown in Figure 9. The band at 1650 cm^−1^ has been assigned to the carbonyl stretching vibration, a number of skeletal ring vibrations at 1598, 1490 and 1413 cm^−1^, the asymmetric stretching of ether group at 1277 and 1190 cm^−1^, the aromatic hydrogen in-plane deformation at 1155, 1215 and 1105 cm^−1^, and the aromatic hydrogen’s out of plane bending modes occurred at 765, 860–841 cm^−1^ (broad band) and diphenyl ketone band at 927 cm^−1^. As the electron beam radiation dose increases, the rate of chain scission increases, which is supported by the fact that the absorption intensity of each group decreased slightly with the increase of the irradiation dose. In addition, the dipole moment of the PEEK polymer changed due to irradiation, whereas infrared scattering of the material surface improved, leading to the weakening of the absorption intensity. Moreover, oxidation related peak at 1730 cm^−1^ enhances as a function of irradiation. On average it can be concluded that an increase in dose by two times leads to further scission and oxidation, which can be seen from the typical radiation-induced mechanical change in PEEK [28,51].

## 4. Discussion

Apparently, the increase in our PEEK surface roughness after irradiation with 2.7 MeV at a dose of 200 kGy is due to electron beam-induced sputtering. A similar effect of polyamide sputtering after irradiation with a 6 MeV pulsed electron beam was observed earlier [52]. In particular, the authors of this work found that the average roughness index (Ra) of the irradiated sample, measured by surface profilometry, increased from 0.06 to 0.1.

The degradation of the mechanical properties of irradiated PEEK can be explained in terms of structural transformations induced by irradiation. The chemical structure of pristine PEEK is illustrated in Figure 10.

The mechanical strength, high thermal and radiation resistance of the polymer is due to the aromatic ring in its composition. Flexibility is provided by the ether bond of oxygen in the main chain. The decrease in density and increase in free volume, leading to greater intermolecular interaction, is provided by the side ketone group. The binding energies of PEEK molecular components highlighted in the literature are shown in Table 5.

In general, the irradiation of polymers including PEEK with high-energy electrons is accompanied by chain scission or crosslinking, as well as changes in their crystallinity and microstructure [34,55]. Obviously, an electron accelerated to an energy of 2.7 MeV can easily break the bonds of any macromolecules in PEEK, since the electron beam energy is much higher than the binding energy of any of the major organic component groups listed in Table 3. We believe that chain (bonds) scission processes were dominant in our experiments on sample irradiation.

We also note that the elastic property measurement results of pristine PEEK obtained from nanoindentation and Brillouin spectroscopy support each other. Further softening of the PEEK structure is deduced to be from the rise of temperature, and irradiation dose.

## 5. Conclusions

The morphology, mechanical and chemical properties of 5 mm thick PEEK plates before and after irradiation with 2.7 MeV electrons with the doses of 200 kGy and 400 kGy were characterized. The CASINO calculation results showed that electrons with an energy of 2.7 MeV penetrated the entire sample. SPM results showed that the surface roughness of PEEK increased when irradiated at a dose of 200 kGy. However, a further increase in the irradiation dose to 400 kGy led to a decrease in the roughness. The nanoindentation and Brillouin results demonstrated with high accuracy, the decay of the hardness and elastic moduli of PEEK with the increase of irradiation dose. PEEK shows more radiation resistance at its surface than in its bulk. Moreover, the elastic properties of both irradiated and pristine PEEK deteriorate with rising temperature.

## Figures and Tables

**Figure 1 polymers-14-03067-f001:**
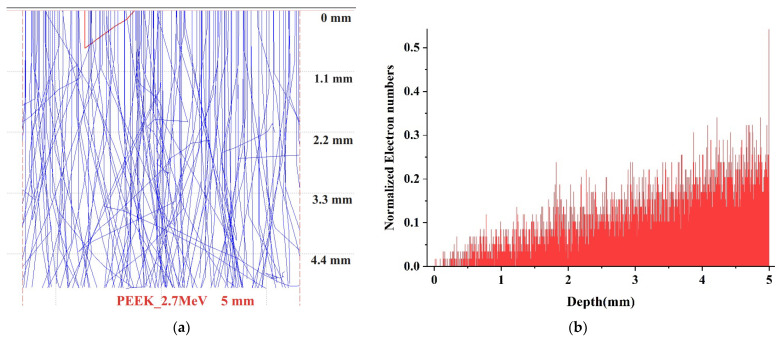
Simulation results of PEEK irradiated with 2.7 MeV electrons: (**a**) Simulation of particle collision process; (**b**) Maximum penetration depth in the sample of the electron trajectories; (**c**) Energy of the transmitted electrons; (**d**) Energy of the backscattered electrons 3.2 Scanning Probe Microscopy.

**Figure 2 polymers-14-03067-f002:**
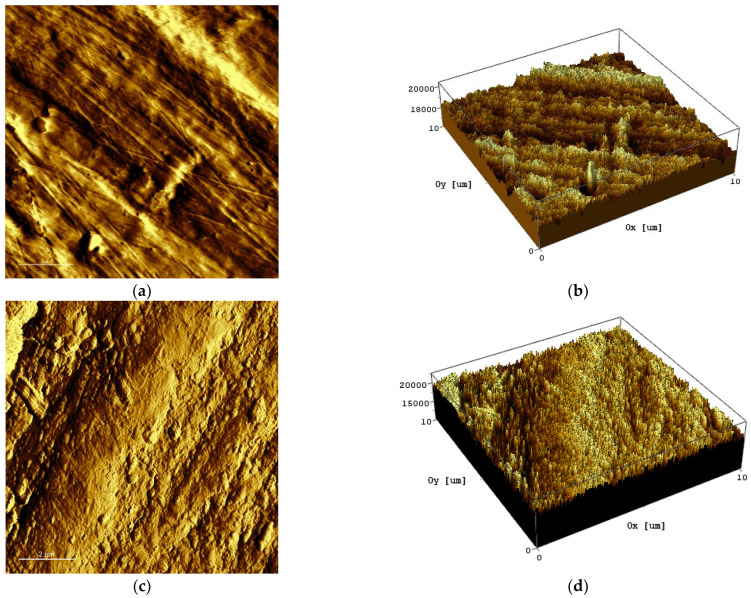
2D and 3D surface morphology of pristine (**a**,**b**) and irradiated PEEK: (**c**,**d**) 200 kGy; (**e**,**f**) 400 kGy.

**Figure 3 polymers-14-03067-f003:**
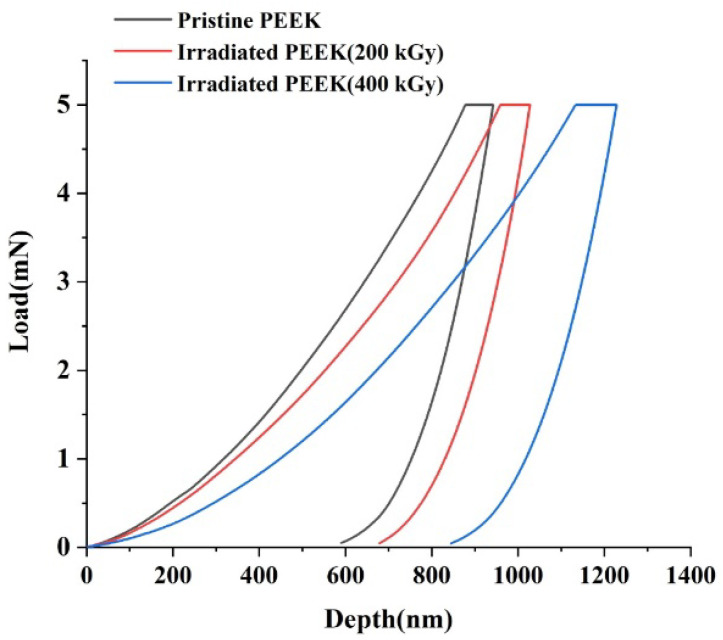
Typical indentation load–displacement curve for the pristine and irradiated PEEK samples at a constant strain rate.

**Figure 4 polymers-14-03067-f004:**
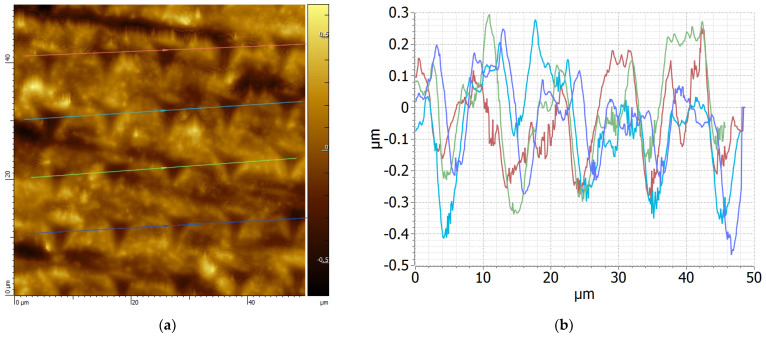
AFM measurements of the surface morphology of a typical PEEK sample after nanoindentation to assess the tip’s penetration depth and volume: (**a**) height measurement, (**b**) assessment of penetration depth.

**Figure 5 polymers-14-03067-f005:**
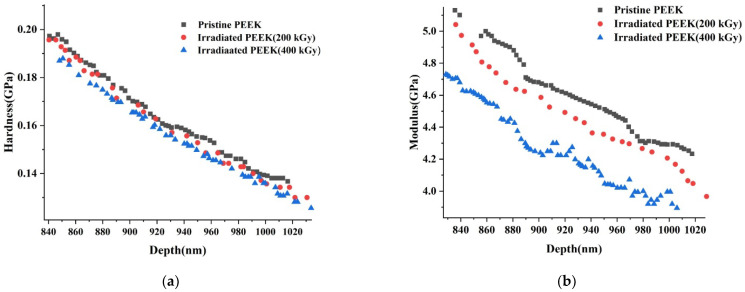
Indentation hardness (**a**) and elastic modulus (**b**) as a function of indentation displacement data for pristine and irradiated PEEK samples.

**Figure 6 polymers-14-03067-f006:**
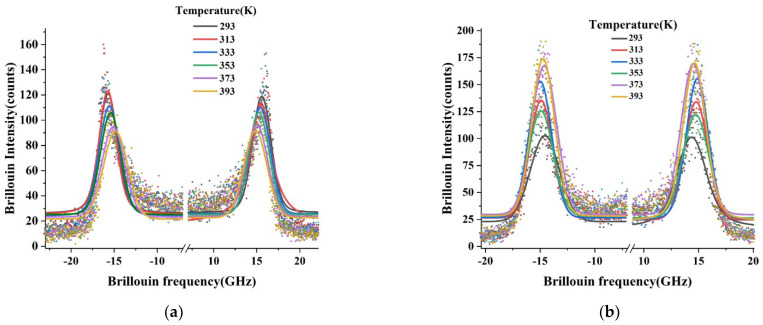
Brillouin spectra of PEEK samples in the 180°-backscattering geometry: (**a**) pristine; (**b**) irradiated.

**Figure 7 polymers-14-03067-f007:**
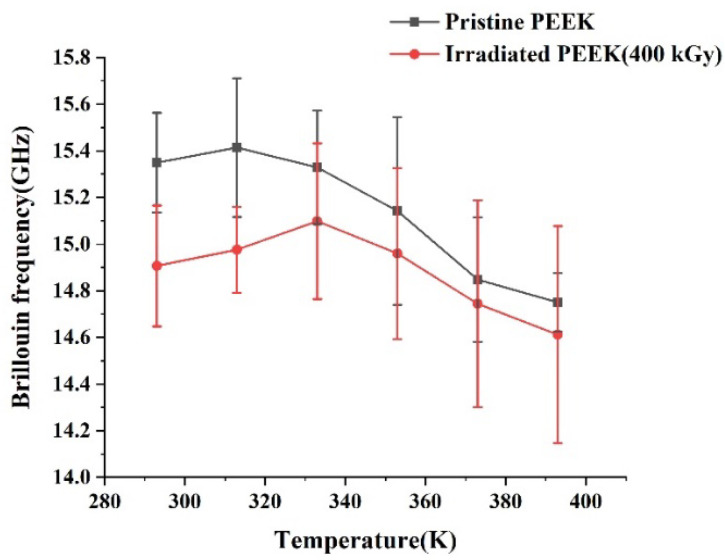
Brillouin shift of pristine and irradiated PEEK samples at different temperatures.

**Figure 8 polymers-14-03067-f008:**
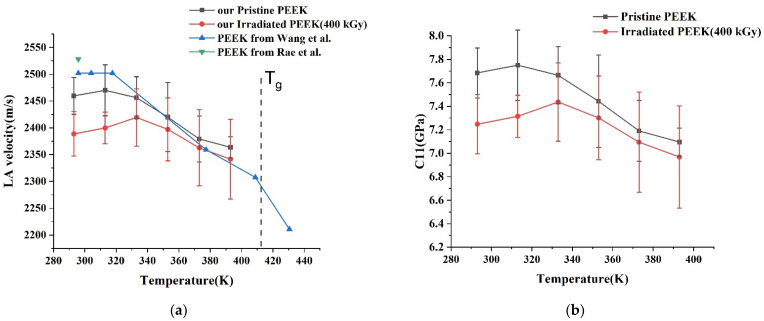
The variation of LA phonon velocity (**a**) and *C*_11_ elastic module (**b**) with temperature.

**Figure 9 polymers-14-03067-f009:**
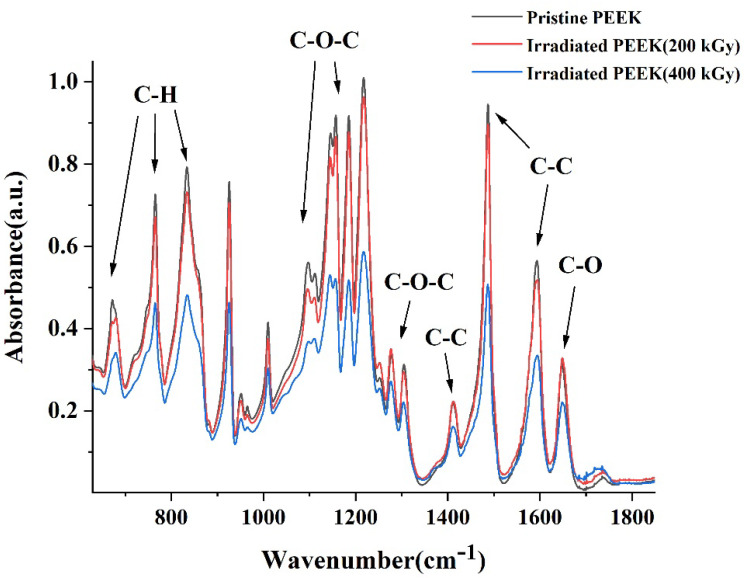
FTIR spectra of pristine and irradiated PEEK.

**Figure 10 polymers-14-03067-f010:**
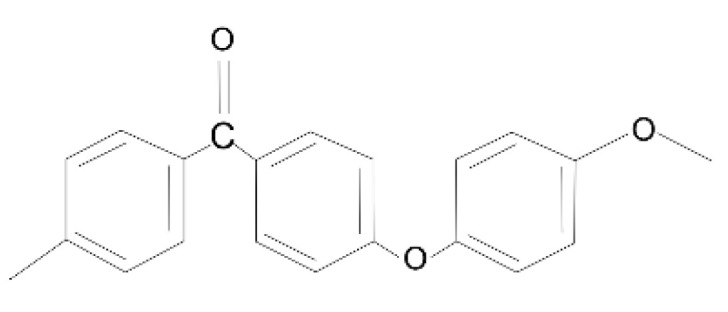
Chemical structure of pristine PEEK.

**Table 1 polymers-14-03067-t001:** Values of roughness parameters of pristine and irradiated PEEK.

PEEK Sample	Ra, [nm]	Rms, [nm]
Pristine	265	339
Irradiated, 200 kGy	750	904
Irradiated, 400 kGy	665	826

**Table 2 polymers-14-03067-t002:** Mechanical properties of PEEK samples measured by nanoindentation.

PEEK Sample	Hardness, [GPa]	Elastic Modulus, [GPa]
Pristine	0.19	4.79
Irradiated, 200 kGy	0.18	4.72
Irradiated, 400 kGy	0.17	4.47

**Table 3 polymers-14-03067-t003:** Berkovich nanoindentation results of pristine PEEK.

Load, [mN]	Depth, [nm]	Hardness, [GPa]	Elastic Modulus, [GPa]	Reference
5	800–1100	0.19	4.79	Our PEEK
	0–1000	0.3	4.7	[47]
	0–5000	0.23	7.8	[48]
1.7–5.2	0–1000	0.25	4.2	[49]
8	0–3500	0.35	6	[50]

**Table 4 polymers-14-03067-t004:** Young’s modulus values of pristine PEEK.

Measurement Techniques	Young’s Modulus, [GPa]
Nanoindentation	4.79
BLS	4.5

**Table 5 polymers-14-03067-t005:** XPS binding energies of PEEK molecular components.

Component	Binding Energy, [eV]	Reference
Ether group, O–C	533.2; 533.4	[53,54]
Carbonyl group, O=C	531.1; 531.27	[53,54]
Aromatic group, C=C	285.0; 284.68	[53,54]
Ether group, C–O	286.6; 286.3	[53,54]
Aromatic group, O, O=C	287.5; 287.29	[53,54]

## Data Availability

The research data are available upon a written request.

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
