# Peer review of "Mechano-Chemical Properties of Electron Beam Irradiated Polyetheretherketone"

_polymers, 2022, doi:10.3390/polym14153067_

Round 1

Reviewer 1 Report

The article does not describe why electron beam radiation was done on PEEK materials. The radiation declined the mechanical properties of the materials. It does not make sense to carry out such research without improving or modifying the properties with significant real-life applications. Instead of lowering the properties, we always want materials with improved or robust mechanical properties for different applications such as medical devices or auto/aero craft parts. In brief, the manuscript does not show any interesting information for publication, and would recommend rejecting this work.

Reviewer 2 Report

Dear Authors,

I have several recommendations; I will go through the chapters of results.

3.1. Assessment of electron penetration depth

You compare two different energies of electrons. However, you use only one energy of electrons for your experiments. Why you use 1 MeV for calculations and not 2 MeV or 0.5 MeV. Please explain this value in some sentences.

3.3 Nanoindentation

Some explanations of the results in table 2 are necessary. From my point of view, only a slight change of hardness take place. However, it exists a bigger difference between the 200 kgy and 400 kgy samples related to the modulus. Why?

Please discuss and explain the results in some sentences.

3.4. Brillouin spectroscopy

In this chapter, you measure only the 400 kgy sample. Why you do not use the 200 kgy?

In addition, in table 3 I found only values for the pristine PEEK without any explanations. Why not 400 kgy sample. Add also some sentences for explanation and assessment of the results.

3.5 FT-IR

I miss a normalization of the spectra. Therefore, it is not possible to compare the intensity of the peaks. Otherwise, the peaks of the pristine sample have the lowest intensity. Please do this and make a new assessment of the results.

BR

The reviewer

Reviewer 3 Report

(lines 49-55) Research methods are only briefly listed. It is necessary to formulate the problem and the purpose of the work.

(line 88) Give explanations for the parameters h and A and link the source of relation (4).

(line 124) Remove parenthesis after Figure 1e.

(line 200) Provide links to the work of Wang and Ray in Figure 8 for PEEK.

(lines 256-263) The conclusions do not fully reflect the results of the work. For example, not specified. that an increase in the irradiation dose to 400 kGy led to a decrease in the surface roughness of the sample.

Reviewer 4 Report

This paper investigated the impact of irradiation on the mechanical and chemical properties of PEEK. I have the following comments that need to be addressed: 

1. Please mention how many replicates of each sample were tested. 

2. Figure 9: Please explain the peak at 1730 cm-1. This peak is absent for pristine sample, and only for irradiated samples. How is this an oxidation peak? FTIR table shows this peak as ketones. This peak should be present for the pristine sample as well. 

3. Can the authors also perform the Tg test to check the change in the glass transition of the irradiated samples? 

Round 2

Reviewer 2 Report

now it is okay

Reviewer 4 Report

The authors have addressed my comments. I will recommend this paper for acceptance.